# Adenylate kinase 1 overexpression increases locomotor activity in medaka fish

**Michiyo Maruyama**[1,2], **Yuko Furukawa**[2], **Masato Kinoshita**[3], **Atsushi Mukaiyama**[4,5], **Shuji Akiyama**[4,5], **Takashi Yoshimura**[1,2]*

1 Laboratory of Animal Integrative Physiology, Graduate School of Bioagricultural Sciences, Nagoya University, Nagoya, Japan, 2 Institute of Transformative Bio-Molecules (WPI-ITbM), Nagoya University, Nagoya, Japan, 3 Division of Applied Biosciences, Graduate School of Agriculture, Kyoto University, Kyoto, Japan, 4 Research Center of Integrative Molecular System (CIMoS), Institute for Molecular Science, National Institute of Natural Sciences, Okazaki, Japan, 5 Department of Functional Molecular Science, SOKENDAI (The Graduate University for Advanced Studies), Okazaki, Japan

* takashiy@agr.nagoya-u.ac.jp

## Abstract

Maintenance of the energy balance is indispensable for cell survival and function. Adenylate kinase (Ak) is a ubiquitous enzyme highly conserved among many organisms. Ak plays an essential role in energy regulation by maintaining adenine nucleotide homeostasis in cells. However, its role at the whole organism level, especially in animal behavior, remains unclear. Here, we established a model using medaka fish (*Oryzias latipes*) to examine the function of Ak in environmental adaptation. Medaka overexpressing the major Ak isoform Ak1 exhibited increased locomotor activity compared to that of the wild type. Interestingly, this increase was temperature dependent. Our findings suggest that cellular energy balance can modulate locomotor activity.

## Introduction

Energy homeostasis is crucial for survival and the maintenance of cell function. Disruption of energy balance is suggested to be associated with many conditions, including obesity, heart failure, and neurodegeneration [1–4]. Adenylate kinase (Ak) is an essential enzyme found in nearly every organism because of its crucial role in cellular energy metabolism [4–7]. Ak catalyzes the nucleotide phosphoryl exchange reaction (2ADP ↔ ATP + AMP) to maintain the adenine nucleotide balance and monitor energy consumption. Ak and its downstream adenine nucleotide signaling pathway have attracted attention because of their vital functions in many biological processes, such as the cell cycle, hormone secretion, and stress tolerance [5,7–11]. To date, nine isoforms of AK (AK1–AK9) have been identified and well characterized in humans [6] (Fig 1A). AK1 is the major isoform of AK and is expressed in the cytosol of most human tissues, with particularly high levels in the brain, the heart, skeletal muscles, and erythrocytes. Several reports have demonstrated the importance of Ak1 in metabolic stress conditions [9–11]. Interestingly, it was recently reported that hyperactive rat strains exhibit increased *Ak1* expression [12]. To our knowledge, however, the role of Ak1 at the whole organism level, especially in animal behavior, remains unclear.

**Data Availability Statement:** All relevant data are within the paper and its Supporting Information files.

**Funding:** This work was supported by a JSPS KAKENHI "Grant-in-Aid for Scientific Research (S)" (19H05643) for T.Y. and the Sasakawa Scientific Research Grant from The Japan Science Society for M.M. There was no additional external funding received for this study. The funders had no role in the study design, data collection and analysis, decision to publish, or preparation of the manuscript.

**Competing interests:** The authors declare no competing interests.

In this study, we sought to clarify the effect of *Ak1* on animal behavioral rhythms. The medaka (*Oryzias latipes*), a small freshwater fish, has recently emerged as a useful vertebrate model because of its small size and short generation time. Additionally, there are well-established methods for transgenic and genomic editing in medaka. Behavioral assays are also well established, for both medaka and zebrafish, but medaka can survive a wider range of water temperatures (4–40˚C) than zebrafish (20–32˚C), and they exhibit seasonal behavioral patterns [13–16]. Therefore, the medaka is an excellent model with which to investigate adaptive strategies to environmental changes.

Here, we successfully established *Ak1*-overexpressing (*Ak1*-OE) medaka. Behavioral assays of *Ak1*-OE medaka larvae revealed a temperature-dependent increase in locomotor activity. These findings shed new light on the function of *Ak1*.

## Materials and methods

### Ethics statement for animal experiments

All animal studies were performed in accordance with the ARRIVE guidelines. All methods were conducted in compliance with the relevant guidelines and regulations and were approved by the Animal Experiment Committee of Nagoya University (approved number: AGR2020009). Animals were euthanized with ethyl 3-aminobenzoate methanesulfonate (MS-222) before sampling. All efforts were made to minimize animal suffering.

### Animals

Medaka fish (*Oryzias latipes*) were obtained from a local dealer (Fuji 3A Project, Nagoya, Japan). The fish were maintained in a housing system (MEITO system; Meito Suien, Nagoya, Japan) under a 14 h light/10 h dark photoperiod (lights on at 05:00 and off at 19:00) at a water temperature of 25˚C. They were fed twice a day with Hikari Labo 450 (KYORIN, Tokyo, Japan).

### Phylogenetic analysis

The phylogenetic tree was constructed using the maximum likelihood method with MEGA 11 (https://www.megasoftware.net/). The GenBank accession numbers of the nucleotide sequences used for the phylogenetic analysis are as follows:

Human (*Homo sapiens*) *AK1*, NM_000476.3; Mouse (*Mus musculus*) *Ak1*, NM_001198790.1; Medaka (*Oryzias latipes*) *Ak1*, XM_004074742.4; Human *AK2*, NM_001199199.3; Mouse *Ak2*, NM_001033966.4; Medaka *Ak2*, XM_011481017.3; Human *AK3*, NM_001199852.2; Mouse *Ak3*, NM_001365071.1; Medaka *Ak3*, XM_004074841.4; Human *AK4*, NM_001005353.3; Mouse *Ak4*, NM_001177602.1; Medaka *Ak4*, XM_004068018.4; Human *AK5*, NM_012093.4, Mouse *Ak5*, NM_001081277.2; Medaka *Ak5*, XM_004078698.4; Human *AK6*, NM_001015891.2; Mouse *Ak6*, NM_027592.3; Medaka *Ak6*, XM_004072821.3; Human *AK7*, NM_001350888.2; Mouse *Ak7*, NM_030187.1; Medaka *Ak7*, XM_011490546.3; Medaka LOC101157414, XM_011492034.2; Human *AK8*, NM_001317958.2; Mouse *Ak8*, NM_001033874.2; Medaka *Ak8*, XM_023958761.1; Human *AK9*, NM_001145128.3; Mouse *Ak9*, NM_001370813.1; Medaka *Ak9*, XM_020714548.2

### Gene expression analysis

Total RNA was purified using an RNeasy micro kit (Qiagen, Hilden, Germany) with DNase I. Extracted RNA was stored at –80˚C. Reverse transcription was performed on total RNA (200 ng) using a ReverTra Ace qPCR RT Master Mix (Toyobo, Osaka, Japan).

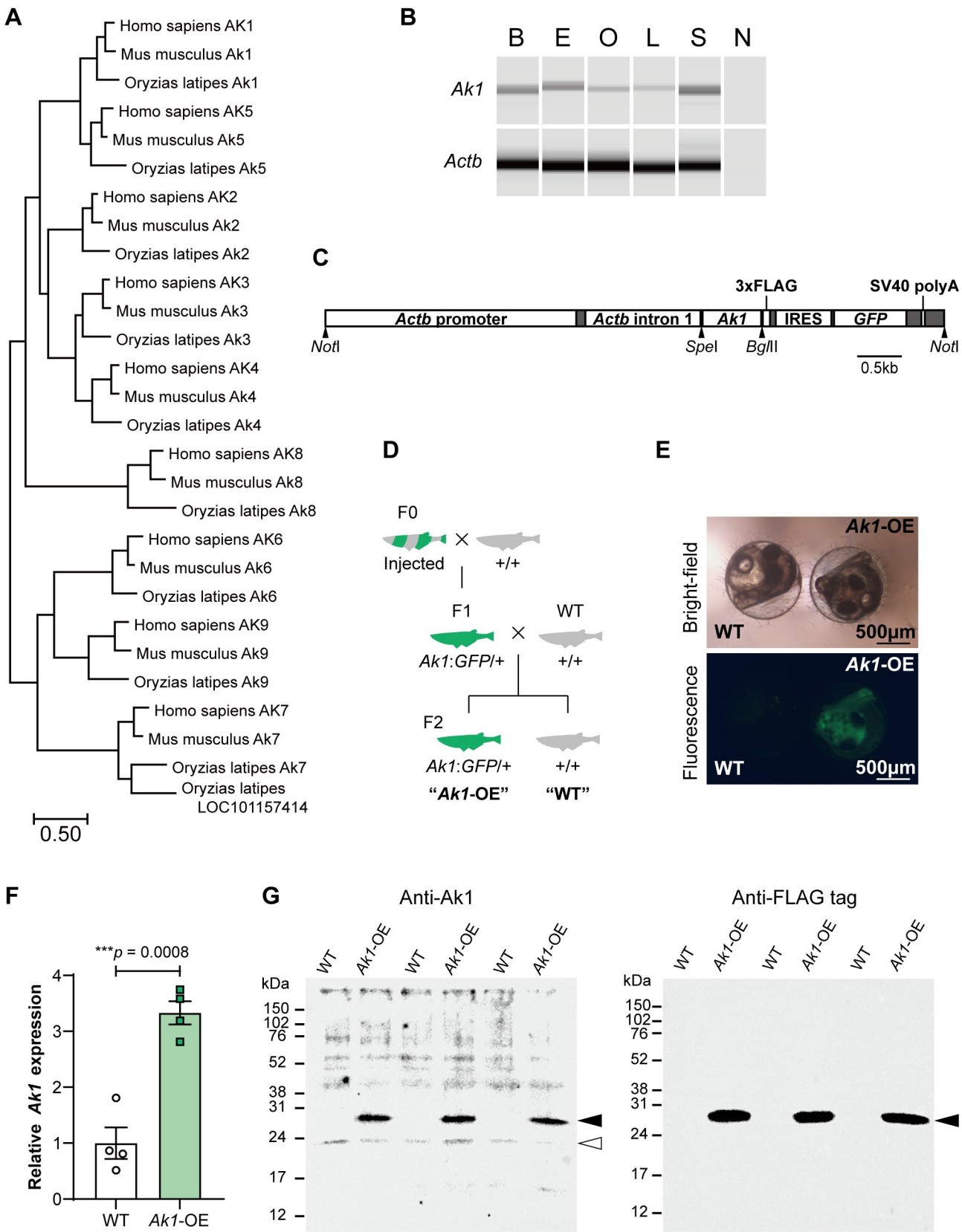

**Fig 1. Generation of *Ak1*-overexpressing medaka.** (A) Phylogenetic tree showing the relationship of the AK isozymes of human, mouse and medaka. (B) Electrophoresis image of *Ak1* and *Actb* RT-PCR products in various adult medaka tissues. B, Brain; E, Eye; O, Ovary; L, Liver; S, Skin; N, No-template control. (C) Structure of *Ak1* overexpression construct. The construct consisted of the medaka *Actb* regulatory region, *Ak1* cDNA (without the stop codon), C-terminus 3× FLAG tag, internal ribosome entry site (IRES), green fluorescent protein (GFP) open reading frame, and SV40 polyA signal. (D) Design of genetic crosses for generating heterozygous *Ak1*-overexpressing (*Ak1*-OE) medaka and their wild-type (WT) siblings. (E) Bright-field and fluorescence images of WT and *Ak1*-OE embryos at 5 days post-fertilization (dpf). (F) *Ak1* expression level of larvae at 9 dpf. Data represent the mean ± SEM, n = 4 each. WT expression level was set at 1. The *p*-value was calculated using two-tailed Welch's *t*-test. Each dot represents an individual value. (G) Western blotting for Ak1 (left) and FLAG tag (right). Black and white arrows indicate Ak1-FLAG and Ak1, respectively.

RT-PCR was performed using TaKaRa Ex Taq (Takara Bio Inc., Shiga, Japan) and the following primers: *Ak1* forward (F): 5'-ACACTCACCTGTCTTCAGGC-3' and reverse (R): 5'-CTGTGTCCAGGGGTACAAGC-3', and *Actb* F: 5'-GATTCCCTTGAAACGAAAAGCC-3' and R: 5'-CAGGGCTGTTGAAAGTCTCAAAC-3'. Amplification was conducted at 94°C for 2 min, followed by 30 cycles of 98°C for 10 s, 60°C for 30 s, 72°C for 30 s and 72°C for 5 min. The PCR products were analyzed using a MultiNA microchip electrophoresis system (Shimadzu Corporation, Kyoto, Japan).

For real-time quantitative PCR (qPCR), 2 μL of the synthesized cDNA was mixed with SYBR Premix Ex Taq II (Takara Bio Inc.) and 0.4 μM primers (same as above) to a total volume of 20 μL. qPCR was performed on a QuantStudio 3 Real-Time PCR System (Applied Biosystems, Waltham, MA, USA). The *Actb* gene was used as an internal control.

## Construction of *Ak1*-OE plasmid

Total RNA was purified from embryos at 4 dpf using an RNeasy micro kit (Qiagen) with DNase I. Reverse transcription of total RNA (200 ng) was performed using a ReverTra Ace qPCR RT kit (Toyobo). The entire *Ak1* open reading frame (excluding the stop codon) was amplified by PCR (98°C for 2 min; 35 cycles of 98°C for 10 s, 65°C for 30 s, 72°C for 1 min, and 72°C for 5 min) using a Q5 Hot Start High-Fidelity 2× Master Mix (New England Biolabs, Ipswich, MA, USA) and primers containing the restriction sites of *Spe*I or *Bgl*II for subsequent subcloning (F: 5'-CCACTAGTATGGCAGACAAAATCAAGGAC-3' and R: 5'- CCAGATCTCTTC AGTGAATCAATAGCCTG-3').

The overexpression plasmid contained the Actin β (*Actb*) promoter followed by the first noncoding exon and the first intron, a 3×FLAG tag, an internal ribosomal entry site (IRES), GFP and SV40 polyA sequences in the *Not*I-*Not*I interval [17]. *Ak1* cDNA and plasmids were digested with *Spe*I and *Bgl*II and purified using NucleoSpin Gel and PCR Clean-up (Macherey-Nagel, Düren, Germany). Ligation was performed using Ligation High Ver. 2 (Toyobo). The sequence of the constructs was confirmed by direct sequencing.

## Generation of *Ak1*-OE medaka

The *Not*I-*Not*I interval of the *Ak1*-OE plasmid was purified and diluted with sterilized water (final concentration: 20 ng/μL) and then microinjected into the cytoplasm of one-cell stage embryos (F0). F0 fish with mosaic GFP fluorescence were raised and crossed with WT fish to obtain heterozygous transgene carriers (F1). F1 fish with GFP fluorescence were crossed with the WT fish. Heterozygous F2 offspring were used for behavioral experiments.

## Western blot analysis

Frozen 9 dpf larvae were homogenized and sonicated in 4× Laemmli sample buffer (Bio-Rad, Hercules, CA, USA). Samples were quantified using Pierce 660 nm Protein Assay Reagent (Thermo Fisher Scientific, Waltham, MA, USA). Twenty-five micrograms of protein per sample were separated by SDS-PAGE and transferred onto a PVDF membrane. The PVDF membrane

was blocked with 5% skim milk in Tris-buffered saline with Tween 20 (TBST) for 1 h at room temperature. Ak1 polyclonal primary antibody (1:500) (14978-1-AP; Proteintech, Rosemont, IL, USA) was dissolved in 5% skim milk in TBST and incubated for 1 h at room temperature. After washing the membrane with TBST, the membranes were incubated with secondary rabbit antibody (1:1000) (NA934-100UL; GE Healthcare, Chicago, IL, USA) dissolved in 5% skim milk in TBST for 1 h at room temperature. The membrane was washed with TBST and incubated with ECL prime (GE Healthcare) for 5 min at room temperature and imaged on a LuminoGraph II EM (ATTO, Tokyo, Japan). After eliminating signals using hydrogen peroxide, the membrane was treated with FLAG (DDDDK) tag primary monoclonal antibody (1:1000) (M185-3S; MBL, Nagoya, Japan) followed by secondary mouse antibody (1:1000) (NA931-100UL; GE Healthcare).

### Behavioral assays

F2 *Ak1*-OE and WT embryos were incubated under a 14L10D cycle at 25˚C until hatching. To measure the locomotor activity, at the end of the light phase at 8 dpf, *Ak1*-OE larvae and their WT siblings were distributed among 48 round well plate (CELL STAR; Greiner Bio-One, Kremsmünster, Austria) each containing 1200 μL of water, with 1 larva per well. The plate was then installed in a DanioVision Observation Chamber (Noldus, Wageningen, the Netherlands). At 19:00 8 dpf, the light was turned off and the larvae were acclimated for 10 h during the dark phase. Behavior was recorded from 05:00 9 dpf to 5:00 12 dpf (for 3 days) under a 14 h light (300 lx) and 10 h dark (0 lx) photoperiod (lights on at 05:00 and off at 19:00) at a water temperature of 15 or 25˚C. Behavior was recorded at 3.75 frames per second and analyzed using the tracking software EthoVision XT (Noldus). Locomotor activity was calculated by measuring the "distance moved" every 10 min.

To examine thigmotaxis, we referred to Schnörr et al. [18]. In brief, larvae were transferred to a 24-well round plate (Nunclon Delta Surface; Thermo Fisher Scientific, Waltham, MA, USA) at 8 dpf. The plate was then installed in a DanioVision Observation Chamber at 9 dpf. After 6 min of acclimation phase (light on), the lights were kept off for 4 min and activities were recorded. The experiments were conducted from 10:00 to 12:00 at 9 dpf. To examine scototaxis and geotaxis, we conducted a light-dark tank test and a novel tank test, respectively [19]. In these assays, adult fish (> 6 months) were used. For the novel tank test, each fish was introduced into a glass tank (100 mm width × 65 mm depth × 142 mm height) filled with water to a height of 100 mm. Recordings were started immediately after transferring the fish and lasted for 7 min. We excluded the first 1 min from the analysis. For the light-dark tank test, a plastic container (150 mm width × 100 mm depth × 55 mm height) was used. The container was divided into two equal-sized sections, colored either white or black. Fish were placed in the light area and allowed to move freely between the dark and light areas for 16 min. We excluded the first 1 min from the analysis.

### Statistical analysis

Data are presented as the mean ± SEM generated using the statistical software GraphPad Prism 9. Two-tailed Welch's *t*-test was used for comparisons between two groups. For comparisons among three or more groups, two-way ANOVA and post hoc tests were conducted using GraphPad Prism 9.

## Results

### Generation of *Ak1*-OE medaka

We first examined *Ak1* expression in various medaka tissues. Reverse transcription polymerase chain reaction (RT-PCR) results showed widespread expression (Fig 1B). To evaluate the effect

of constitutively high *Ak1* expression, we used a construct that overexpressed *Ak1* in response to regulation of the *Actb* promoter (Fig 1C). This construct enabled us to visualize transgene expression via green fluorescent protein (GFP), which was linked to the 3' internal ribosome entry site (IRES) downstream of *Actb-Ak1* [17].

To generate *Ak1*-OE medaka, the construct was microinjected into the cytoplasm of one-cell stage embryos. Exhibiting chimeric GFP expression embryos were raised as F0 medaka. These medaka were paired with the wild type to obtain heterozygous transgene carriers (F1). GFP-expressing F1 medaka were then crossed with the wild type to generate heterozygous F2 offspring (Fig 1D and 1E). Overexpression of *Ak1* in GFP-positive F2 medaka was confirmed by quantitative RT-PCR (qPCR) (Fig 1F). We also determined Ak1 protein levels in both *Ak1*-OE and WT by western blot analysis using the Ak1 antibody (Fig 1G, left). In both *Ak1*-OE and WT samples, we observed bands close to the expected band size of medaka Ak1 (21.4 kDa). In addition, we found strong bands at high molecular weights only in *Ak1*-OE samples. In our overexpression construct, we added a 3× FLAG tag downstream of the Ak1 sequence. Therefore, we expected that the strong bands observed were FLAG-tagged Ak1. To confirm this, we examined western blot analysis using a FLAG-tag antibody. As expected, we observed bands at the same position as those detected with the Ak1 antibody (Fig 1G, right). From these results, we confirmed Ak1 overexpression at both the gene expression and protein levels.

### *Ak1*-OE larvae exhibited increased locomotor activity at 25˚C

The high-throughput behavioral tracking assay is well established in larvae of zebrafish and medaka fish [14,15]. Therefore, we used this method to examine the locomotor activity of medaka larvae. *Ak1*-OE embryos and their wild-type (WT) siblings were raised in an incubator under a 14 h light/10 h dark (14L10D) cycle (lights on at 05:00 and off at 19:00) at 25˚C. At the end of the light phase at 8 days post-fertilization (dpf), hatched larvae were placed in each well of a 48-well plate. The plate was then put into a high-throughput behavior tracking system. Larvae were maintained under a 14L10D cycle at 25˚C. After 10 h of habituation, at the onset of the light phase at 9 dpf, locomotor activity was tracked for three full days (Fig 2A). The tracking profiles showed that both *Ak1*-OE and WT larvae were more active during the light phase than during the dark phase (Fig 2B). Compared to that on the first day of the assay, the total distance moved by the WT larvae significantly decreased on the third day (Fig 2C). As in previous studies [14,15], larvae were not fed during the assay to avoid interrupting it. This may have caused decreased energy, and hence, decreased activity, later in the experiment. Notably, the average locomotor activity of the *Ak1*-OE larvae tended to be higher than that of the WT larvae throughout the assay. This trend was particularly obvious on day 3, at which point the total locomotor activity of the *Ak1*-OE larvae was significantly higher than that of the WT larvae, both in the light and dark phases (Fig 2D).

To test whether the increase in locomotor activity of *Ak1*-OE was due to changes in some taxis, we conducted several additional behavioral tests. When we examined changes in thigmotaxis between *Ak1*-OE and WT as described by Schnörr et al. [18], no difference was observed (S1A and S1B Fig). We also conducted a light-dark tank test (S1C and S1D Fig) and a novel tank test (S1E and S1F Fig) for scototaxis and geotaxis, respectively. However, we found no difference between *Ak1*-OE and WT in these tests, suggesting that Ak1 indeed affects locomotor activity but not taxis.

### *Ak1*-OE larvae did not exhibit increased locomotor activity at 15˚C

Ambient temperature directly affects the metabolic rate of ectothermic animals. Therefore, we further examined the locomotor activity of *Ak1*-OE larvae at a lower temperature (15˚C), a

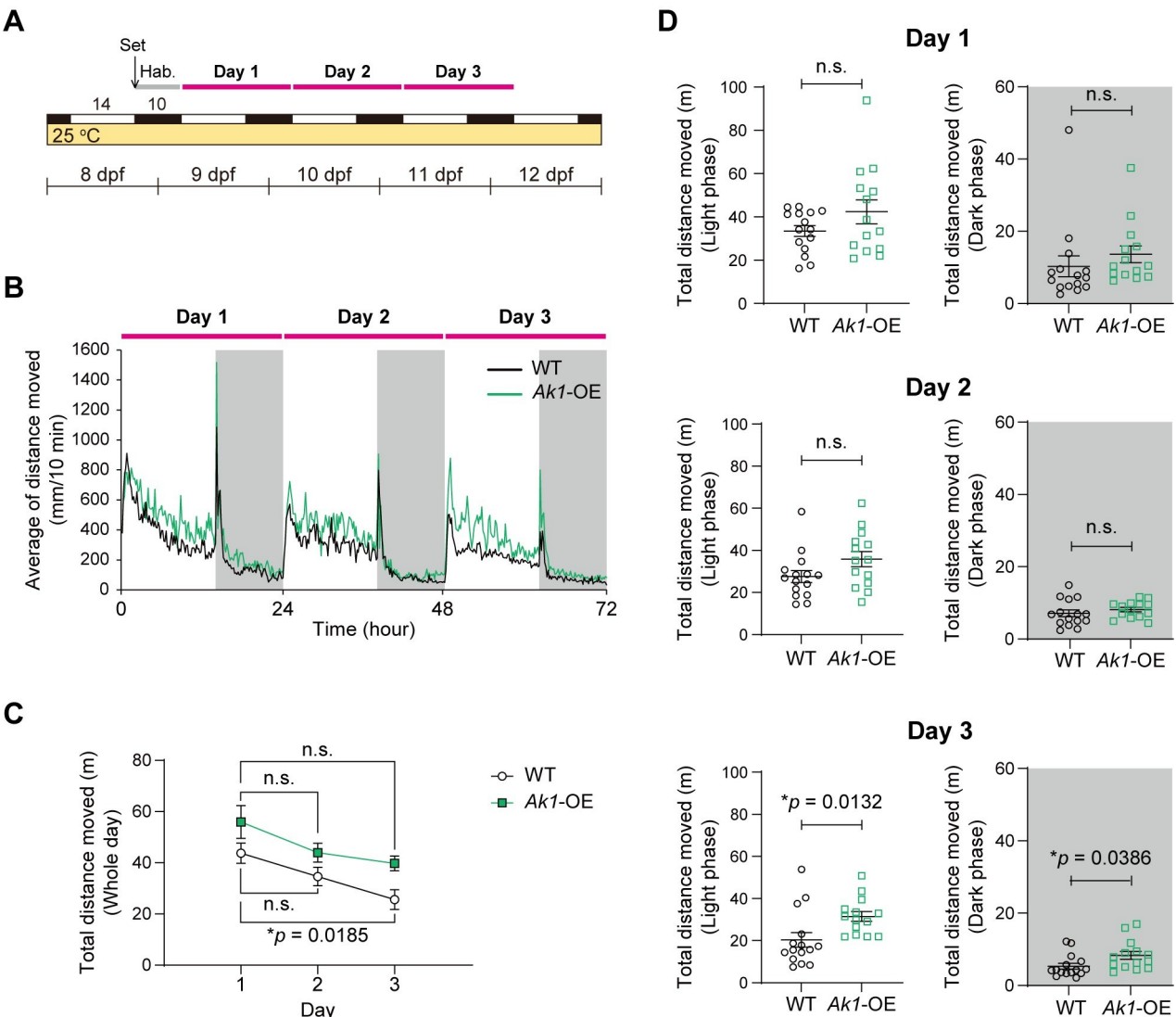

**Fig 2. Locomotor activity of *Ak1*-overexpressing larvae at 25˚C.** (A) Structure of the behavioral assay. Larvae (8 dpf) were placed into a photoperiod- and temperature-controlled chamber (14L10D, 25˚C) at the end of the light phase. After 10 h of habituation (Hab.), recording was started at the onset of the light phase at 9 dpf and continued for 3 full days. (B) Mean locomotor activity of *Ak1*-overexpressing (*Ak1*-OE) and wild-type (WT) larvae. (C) Total locomotor activity of *Ak1*-OE and WT larvae per day. Data represent the mean ± SEM and were analyzed via two-way repeated measures ANOVA. Effect of time, $F_{(1.386, 37.43)} = 11.45$, $p = 0.0006$; effect of overexpression, $F_{(1, 27)} = 7.891$, $p = 0.0091$; effect of interaction, $F_{(2, 54)} = 0.2186$, $p = 0.8044$. *$p < 0.05$; n.s., not significant (Dunnett's multiple comparisons test, vs. day 1). (D) Total locomotor activity of *Ak1*-OE and WT larvae per day during the light phase (white) and dark phase (gray). Data represent the mean ± SEM. The *p*-values (*$p < 0.05$) were calculated using two-tailed Welch's *t*-test. Each dot represents an individual value. (B–D) n = 15 (WT), n = 14 (*Ak1*-OE).

condition associated with lower energy expenditure. After they were raised at 25˚C, at 8 dpf, larvae were placed in a tracking chamber and exposed to a 14L10D cycle at 15˚C (Fig 3A). In both the *Ak1*-OE and WT larvae, locomotor activity levels on days 2 and 3 were similar to those on day 1 (Fig 3B and 3C). This suggests that the larvae conserved their energy until the end of the experiment at the lower temperature. Unlike the results at 25˚C, there was no difference in locomotor activity between the *Ak1*-OE and WT larvae throughout the experiment (Fig 3D). This suggests that the increase in the locomotor activity of *Ak1*-OE larvae was temperature dependent.

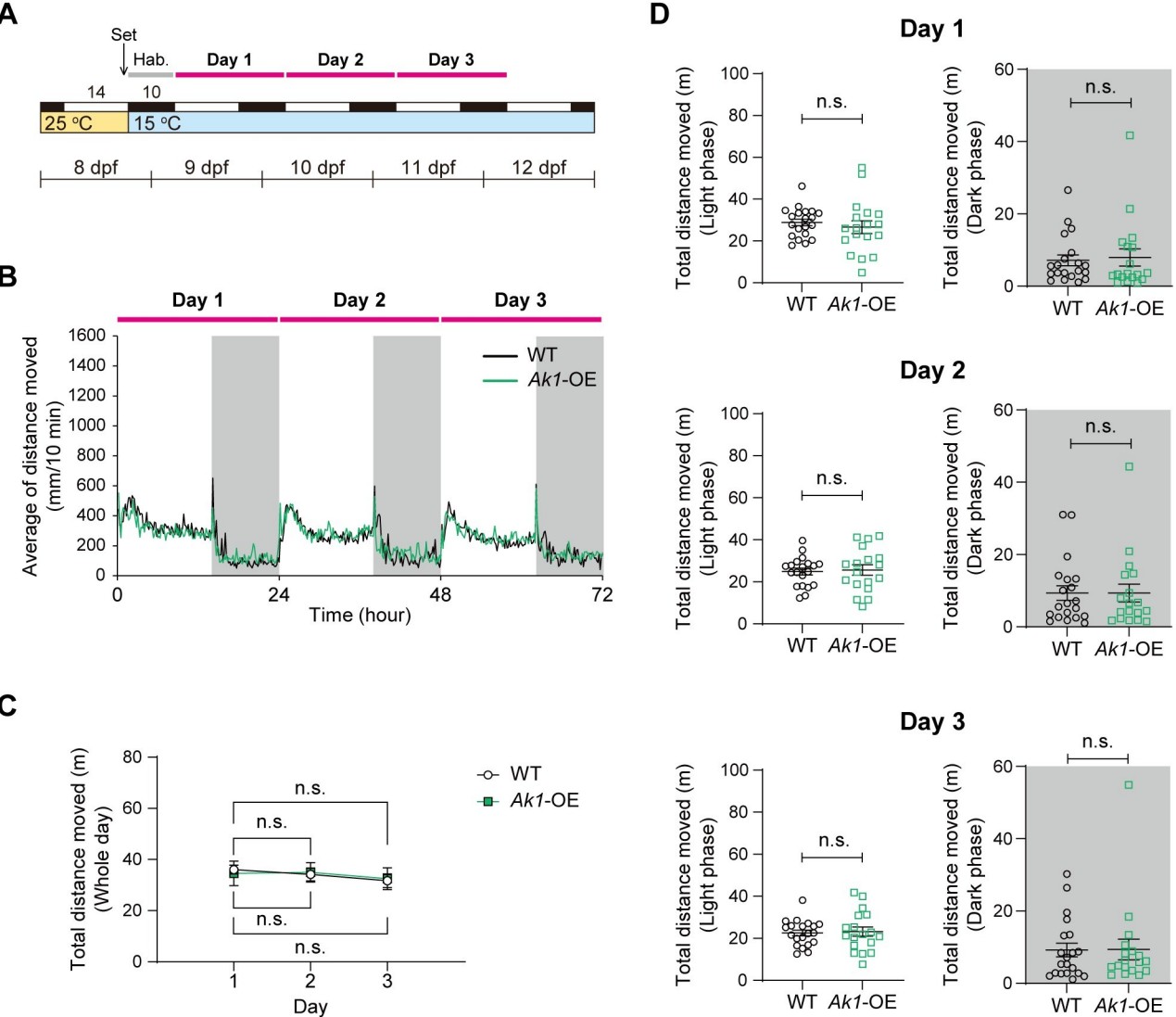

**Fig 3. Locomotor activity of *Ak1*-overexpressing larvae at 15°C.** (A) Structure of the behavioral assay. Larvae were maintained in a chamber under a 14L10D cycle at 15°C. (B) Mean locomotor activity of *Ak1*-overexpressing (*Ak1*-OE) and wild-type (WT) larvae. (C) Total locomotor activity of *Ak1*-OE and WT larvae per day. Data represent the mean ± SEM and were analyzed via two-way repeated measures ANOVA. Effect of time, $F_{(1.910, 68.74)}$ = 1.848, $p$ = 0.1669; effect of overexpression, $F_{(1, 36)}$ = 3.607×10$^{-5}$, $p$ = 0.9952; effect of interaction, $F_{(2, 72)}$ = 0.2813, $p$ = 0.7556. n.s., not significant (Dunnett's multiple comparisons test, vs. day 1). (D) Total locomotor activity of *Ak1*-OE and WT larvae per day during the light phase (white) and dark phase (gray). Data represent the mean ± SEM. n.s., not significant (two-tailed Welch's *t*-test). Each dot represents an individual value. (B–D) n = 20 (WT), n = 18 (*Ak1*-OE).

## Discussion

Ak are essential enzymes that play critical roles in metabolic monitoring and signaling of cells. Among the Ak family members, we focused on Ak1, the major cytosolic isoform of Ak. In the present study, in various medaka tissues, we observed *Ak1* expression similar to that reported in other animals [6] (Fig 1B). Ak1 is highly conserved among many organisms. Indeed, the alignment of human AK1 (NP_000467.1, 194 aa) with medaka Ak1 (XP_004074790.1, 194 aa) exhibited 75% identical residues and 88% similar residues (via the NCBI BLAST Needleman-Wunsch Global Align function).

To know the function of Ak1 at the whole organism level, we established an *Ak1*-OE medaka line (Fig 1C–1G). Behavioral assays of the medaka larvae revealed that *Ak1*-OE larvae exhibited increased locomotor activity compared with that of the WT larvae at 25˚C, and the difference between the *Ak1*-OE and WT larvae was most obvious on the last day of the experiment (Fig 2D). *Ak1*-OE larvae maintained a higher activity level throughout the entire experiment, whereas the total amount of locomotor activity of the WT larvae significantly decreased on day 3 (Fig 2C). In contrast, throughout the experiment at 15˚C, the locomotor activity of *Ak1*-OE larvae did not differ from that of the WT larvae (Fig 3D). Several knockout studies have shown that *Ak1* deficiency can be compensated for by other members of the Ak family under normal conditions, whereas *Ak1* deficiency leads to failure of metabolic homeostasis under stress conditions [10,11,20]. As our behavioral assays were conducted without feeding, the lack of energy later in the experiment might have mimicked metabolic stress even at the moderate temperature of 25˚C, and this could have caused the large difference in activity between the *Ak1*-OE and WT larvae on the last day of the assay.

In summary, Ak1 is known to play crucial roles in cell function. Our results indicate the importance of *Ak1* not only at the cellular level but also at the behavioral level.

## Supporting information

**S1 Fig. Comparison of thigmotaxis, scototaxis, and geotaxis between *Ak1*-OE and WT.** (A) Schematic drawing of thigmotaxis measurement test. (B) Results of thigmotaxis measurement test (n = 11 each). (C) Schematic of light-dark tank test. (D) Results of light-dark tank test (n = 8 each). (E) Schematic representation of the novel tank test. (F) Results of the novel tank test (n = 8 each). All data are presented as the mean ± SEM. The *p*-value was calculated using a two-tailed Welch's *t*-test. Each dot represents an individual value.
(TIF)

**S1 Raw images.**
(PDF)

## Acknowledgments

We thank Drs. T. Nishiwaki-Ohkawa, A. Shinomiya and T. Nakayama for helpful discussions. We also thank A. Ieda and A. Matsumiya for technical assistance.

## Author Contributions

**Conceptualization:** Michiyo Maruyama, Takashi Yoshimura.

**Funding acquisition:** Michiyo Maruyama, Takashi Yoshimura.

**Investigation:** Michiyo Maruyama, Yuko Furukawa.

**Methodology:** Masato Kinoshita, Atsushi Mukaiyama, Shuji Akiyama.

**Resources:** Masato Kinoshita, Takashi Yoshimura.

**Supervision:** Atsushi Mukaiyama, Shuji Akiyama, Takashi Yoshimura.

**Writing – original draft:** Michiyo Maruyama.

**Writing – review & editing:** Takashi Yoshimura.

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
