## [Decision Letter · Decision Letter 0]

27 Oct 2021

PONE-D-21-29550Adenylate kinase 1 overexpression increases locomotor activity in medaka fishPLOS ONE

Dear Dr. Yoshimura,

Thank you for submitting your manuscript to PLOS ONE. After careful consideration, we feel that it has merit but does not fully meet PLOS ONE’s publication criteria as it currently stands. Therefore, we invite you to submit a revised version of the manuscript that addresses the points raised during the review process.

Your manuscript has been evaluated by two reviewers who are recognized experts in this field. Both reviewers made recommendations that you should consider in order to carefully revise your manuscript.

We look forward to receiving your revised manuscript.

Kind regards,

Hubert Vaudry

Academic Editor

PLOS ONE

Journal Requirements:

2. To comply with PLOS ONE submissions requirements, in your Methods section, please provide additional information on the animal research and ensure you have included details on (1) methods of sacrifice, (2) methods of anesthesia and/or analgesia, and (3) efforts to alleviate suffering.

 [This work was supported in part by a JSPS KAKENHI “Grant-in-Aid for Scientific Research (S)” (19H05643) for T.Y. and the Sasakawa Scientific Research Grant from The Japan Science Society for M.M.]

Reviewers' comments:

Reviewer's Responses to Questions

**Comments to the Author**

1. Is the manuscript technically sound, and do the data support the conclusions?

Reviewer #1: Yes

Reviewer #2: Yes

2. Has the statistical analysis been performed appropriately and rigorously? 

Reviewer #1: I Don't Know

Reviewer #2: Yes

3. Have the authors made all data underlying the findings in their manuscript fully available?

Reviewer #1: Yes

Reviewer #2: Yes

4. Is the manuscript presented in an intelligible fashion and written in standard English?

Reviewer #1: Yes

Reviewer #2: Yes

5. Review Comments to the Author

Reviewer #1: This manuscript by Maruyama and colleagues addresses an interesting issue regarding the links between cellular energy metabolism and behavioural activity. Using a medaka transgenic model, the authors study the consequences of overexpression of one key isoform of the enzyme Adenylate kinase (AK1) on locomotor activity of larval medaka following Day 8 post fertilization. The overall basal level of locomotor activity measured in the transgenics appears to be significantly higher than in the wild type controls, but only when measured at 25°C. Instead, during the locomotor activity assay performed at 15°C, activity of the wild type and transgenics were indistinguishable.

This is an interesting observation that deserves to be published, however I would propose the following modifications:

1) It is well know that in teleosts, many gene families are larger than in mammals due to the effect of extra gene copies resulting from ancestral genome duplication events. This leads to the obvious question: are there only 9 isoforms of AK in medaka? If there are more, it would be helpful to include a figure with a phylogenetic tree comparing the mammalian and medaka situation.

2) While it is clear that the AK1 transgene is expressed from a strong, widely expressed promoter, there is no evidence provided comparing AK1 protein levels or AK1 enzymatic activity in the transgenics and wild types. This would be particularly important information to fully interpret the comparison of the behaviour at the two different temperatures. If technically feasible, such extra information would be invaluable for interpreting the results.

Reviewer #2: This manuscript by Maruyama et al. describes the effect of adenylate kinase 1 overexpression on locomotor activity in medaka. The authors found that Ak1-OE increases locomotor activity at a water temperature of 25C in the light period. As such a change did not occur at 15C, the authors suggested the possibility of modulating locomotor activity by Ak1 depending water temperature. This MS was short and well written about well-designed experiments. I have only one question, and changes in locomotor activity may reflect some change in taxis. Did the authors observe any changes in thigmotaxis, scototaxis, geotaxis, etc.

6. PLOS authors have the option to publish the peer review history of their article (what does this mean?). If published, this will include your full peer review and any attached files.

Reviewer #1: No

Reviewer #2: No

---

## [Author Response · Author response to Decision Letter 0]

3 Dec 2021

Point-by-point responses to the reviewers’ comments to Maruyama et al., “Adenylate kinase 1 overexpression increases locomotor activity in medaka fish”

We are grateful to the reviewers for their positive and valuable comments, which have helped us improve our manuscript. The reviewers’ comments are shown in blue font below, and our responses are shown in black font.

Reviewer #1: This manuscript by Maruyama and colleagues addresses an interesting issue regarding the links between cellular energy metabolism and behavioural activity. Using a medaka transgenic model, the authors study the consequences of overexpression of one key isoform of the enzyme Adenylate kinase (AK1) on locomotor activity of larval medaka following Day 8 post fertilization. The overall basal level of locomotor activity measured in the transgenics appears to be significantly higher than in the wild type controls, but only when measured at 25°C. Instead, during the locomotor activity assay performed at 15°C, activity of the wild type and transgenics were indistinguishable.

This is an interesting observation that deserves to be published, however I would propose the following modifications:

1) It is well known that in teleosts, many gene families are larger than in mammals due to the effect of extra gene copies resulting from ancestral genome duplication events. This leads to the obvious question: are there only 9 isoforms of AK in medaka? If there are more, it would be helpful to include a figure with a phylogenetic tree comparing the mammalian and medaka situation.

Response: Only Ak7 was duplicated in medaka. As suggested, we have added the phylogenetic tree in Fig. 1A. Accordingly, we have changed the figure order.

We have also added the following text in the Methods section:

Phylogenetic analysis

The phylogenetic tree was constructed using the maximum likelihood method with MEGA 11 (https://www.megasoftware.net/). The GenBank accession numbers of the nucleotide sequences used for the phylogenetic analysis are as follows:

Human (Homo sapiens) AK1, NM_000476.3; Mouse (Mus musculus) Ak1, NM_001198790.1; Medaka (Oryzias latipes) Ak1, XM_004074742.4; Human AK2, NM_001199199.3; Mouse Ak2, NM_001033966.4; Medaka Ak2, XM_011481017.3; Human AK3, NM_001199852.2; Mouse Ak3, NM_001365071.1; Medaka Ak3, XM_004074841.4; Human AK4, NM_001005353.3; Mouse Ak4, NM_001177602.1; Medaka Ak4, XM_004068018.4; Human AK5, NM_012093.4, Mouse Ak5, NM_001081277.2; Medaka Ak5, XM_004078698.4; Human AK6, NM_001015891.2; Mouse Ak6, NM_027592.3; Medaka Ak6, XM_004072821.3; Human AK7, NM_001350888.2; Mouse Ak7, NM_030187.1; Medaka Ak7, XM_011490546.3; Medaka LOC101157414, XM_011492034.2; Human AK8, NM_001317958.2; Mouse Ak8, NM_001033874.2; Medaka Ak8, XM_023958761.1; Human AK9, NM_001145128.3; Mouse Ak9, NM_001370813.1; Medaka Ak9, XM_020714548.2

2) While it is clear that the AK1 transgene is expressed from a strong, widely expressed promoter, there is no evidence provided comparing AK1 protein levels or AK1 enzymatic activity in the transgenics and wild types. This would be particularly important information to fully interpret the comparison of the behaviour at the two different temperatures. If technically feasible, such extra information would be invaluable for interpreting the results.

Response: We determined Ak1 protein levels in both Ak1-OE and WT by western blot analysis using the Ak1 antibody.

In both Ak1-OE and WT samples, we observed bands close to the expected band size of medaka Ak1 (21.4 kDa). In addition, we found strong bands at high molecular weights only in Ak1-OE samples. In our overexpression construct, we added a 3× FLAG tag downstream of the Ak1 sequence. Therefore, we expected that the strong bands observed were FLAG-tagged Ak1. To confirm this, we conducted western blot analysis using a FLAG-tag antibody. As expected, we observed bands at the same position as those detected with the Ak1 antibody. From these results, we confirmed Ak1 overexpression at both the gene expression and protein levels. We have added these results in the revised manuscript.

We have also added the following text in the Methods section:

Western blot analysis

Frozen 9 dpf larvae were homogenized and sonicated in 4× Laemmli sample buffer (Bio-Rad, Hercules, CA, USA). Samples were quantified using Pierce 660 nm Protein Assay Reagent (Thermo Fisher Scientific, Waltham, MA, USA). Twenty-five micrograms of protein per sample were separated by SDS-PAGE and transferred onto a PVDF membrane. The PVDF membrane was blocked with 5% skim milk in Tris-buffered saline with Tween 20 (TBST) for 1 h at room temperature. Ak1 polyclonal primary antibody (1:500) (14978-1-AP; Proteintech, Rosemont, IL, USA) was dissolved in 5% skim milk in TBST and incubated for 1 h at room temperature. After washing the membrane with TBST, the membranes were incubated with secondary rabbit antibody (1:1000) (NA934-100UL; GE Healthcare, Chicago, IL, USA) dissolved in 5 % skim milk in TBST for 1 h at room temperature. The membrane was washed with TBST and incubated with ECL prime (GE Healthcare) for 5 min at room temperature and imaged on a LuminoGraph II EM (ATTO, Tokyo, Japan). After eliminating signals using hydrogen peroxide, the membrane was treated with FLAG (DDDDK) tag primary monoclonal antibody (1:1000) (M185-3S; MBL, Nagoya, Japan) followed by secondary mouse antibody (1:1000) (NA931-100UL; GE Healthcare).

Reviewer #2: This manuscript by Maruyama et al. describes the effect of adenylate kinase 1 overexpression on locomotor activity in medaka. The authors found that Ak1-OE increases locomotor activity at a water temperature of 25C in the light period. As such a change did not occur at 15C, the authors suggested the possibility of modulating locomotor activity by Ak1 depending water temperature. This MS was short and well written about well-designed experiments. I have only one question, and changes in locomotor activity may reflect some change in taxis. Did the authors observe any changes in thigmotaxis, scototaxis, geotaxis, etc.

Response: To test whether the increase in locomotor activity of Ak1-OE was due to changes in some taxis, we conducted several additional behavioral tests. When we examined changes in thigmotaxis between Ak1-OE and WT as described by Schnorr et al. [18], no difference was observed. We also conducted a light-dark tank test and a novel tank test for scototaxis and geotaxis, respectively. However, we found no difference between Ak1-OE and WT in these tests, suggesting that Ak1 indeed affects locomotor activity but not taxis. We have added these results to S1 Fig: Comparison of thigmotaxis, scototaxis, and geotaxis between Ak1-OE and WT.

(A) Schematic drawing of thigmotaxis measurement test. (B) Results of thigmotaxis measurement test (n = 11 each). (C) Schematic of light-dark tank test. (D) Results of light-dark tank test (n = 8 each). (E) Schematic representation of the novel tank test. (F) Results of the novel tank test (n = 8 each). All data are presented as the mean ± SEM. The p-value was calculated using a two-tailed Welch’s t-test. Each dot represents an individual value.

We have also added the following text in the Methods section:

To examine thigmotaxis, we referred to Schnorr et al. [18]. In brief, larvae were transferred to a 24-well round plate (Nunclon Delta Surface; Thermo Fisher Scientific, Waltham, MA, USA) at 8 dpf. The plate was then installed in a DanioVision Observation Chamber at 9 dpf. After 6 min of acclimation phase (light on), the lights were kept off for 4 min and activities were recorded. The experiments were conducted from 10:00 to 12:00 at 9 dpf. 

To examine scototaxis and geotaxis, we conducted a light-dark tank test and a novel tank test, respectively [19]. In these assays, adult fish (> 6 months) were used. For the novel tank test, each fish was introduced into a glass tank (100 mm width × 65 mm depth × 142 mm height) filled with water to a height of 100 mm. Recordings were started immediately after transferring the fish and lasted for 7 min. We excluded the first 1 min from the analysis. For the light-dark tank test, a plastic container (150 mm width × 100 mm depth × 55 mm height) was used. The container was divided into two equal-sized sections, colored either white or black. Fish were placed in the light area and allowed to move freely between the dark and light areas for 16 min. We excluded the first 1 min from the analysis.

Response to Academic Editor:

Response: We have revised the manuscript according to the guidelines of PLOS ONE as suggested.

2. To comply with PLOS ONE submissions requirements, in your Methods section, please provide additional information on the animal research and ensure you have included details on (1) methods of sacrifice, (2) methods of anesthesia and/or analgesia, and (3) efforts to alleviate suffering.

Response: We added the following text on “Ethics statement for animal experiments” in the Methods section:

Animals were euthanized with ethyl 3-aminobenzoate methanesulfonate (MS-222) before sampling. All efforts were made to minimize animal suffering.

 [This work was supported in part by a JSPS KAKENHI “Grant-in-Aid for Scientific Research (S)” (19H05643) for T.Y. and the Sasakawa Scientific Research Grant from The Japan Science Society for M.M.]

Response: We have changed the original statement to:

This work was supported by a JSPS KAKENHI “Grant-in-Aid for Scientific Research (S)” (19H05643) for T.Y. and the Sasakawa Scientific Research Grant from The Japan Science Society for M.M. There was no additional external funding received for this study. The funders had no role in the study design, data collection and analysis, decision to publish, or preparation of the manuscript.

Response: We have amended as follows:

All relevant data are within the manuscript and its Supporting Information files.

Response: We have uploaded the blot/gel image file (S1_raw_images) as a Supporting Information file.

---

## [Decision Letter · Decision Letter 1]

23 Dec 2021

Adenylate kinase 1 overexpression increases locomotor activity in medaka fish

PONE-D-21-29550R1

Dear Dr. Yoshimura,

We’re pleased to inform you that your manuscript has been judged scientifically suitable for publication and will be formally accepted for publication once it meets all outstanding technical requirements.

Kind regards,

Hubert Vaudry

Academic Editor

PLOS ONE

Additional Editor Comments (optional):

Reviewers' comments:

Reviewer's Responses to Questions

**Comments to the Author**

1. If the authors have adequately addressed your comments raised in a previous round of review and you feel that this manuscript is now acceptable for publication, you may indicate that here to bypass the “Comments to the Author” section, enter your conflict of interest statement in the “Confidential to Editor” section, and submit your "Accept" recommendation.

Reviewer #1: All comments have been addressed

2. Is the manuscript technically sound, and do the data support the conclusions?

Reviewer #1: Yes

3. Has the statistical analysis been performed appropriately and rigorously? 

Reviewer #1: I Don't Know

4. Have the authors made all data underlying the findings in their manuscript fully available?

Reviewer #1: Yes

5. Is the manuscript presented in an intelligible fashion and written in standard English?

Reviewer #1: Yes

6. Review Comments to the Author

Reviewer #1: (No Response)

7. PLOS authors have the option to publish the peer review history of their article (what does this mean?). If published, this will include your full peer review and any attached files.

Reviewer #1: **Yes: **Nicholas S. Foulkes

---

## [Editor Report · Acceptance letter]

26 Dec 2021

PONE-D-21-29550R1 

Adenylate kinase 1 overexpression increases locomotor activity in medaka fish 

Dear Dr. Yoshimura:

I'm pleased to inform you that your manuscript has been deemed suitable for publication in PLOS ONE. Congratulations! Your manuscript is now with our production department. 

Kind regards, 

on behalf of

Dr. Hubert Vaudry 

Academic Editor

PLOS ONE